# Employees’ Appraisals and Trust of Artificial Intelligences’ Transparency and Opacity

**DOI:** 10.3390/bs13040344

**Published:** 2023-04-20

**Authors:** Liangru Yu, Yi Li, Fan Fan

**Affiliations:** 1School of Economics and Management, Chongqing University of Posts and Telecommunications, Chongqing 400065, China; 2Faculty of Collaborative Regional Innovation, Ehime University, Matsuyama 790-8566, Ehime, Japan

**Keywords:** AI transparency trust in, challenge appraisals, threat appraisals, employees’ trust in AI, domain knowledge about AI

## Abstract

Artificial intelligence (AI) is being increasingly used as a decision agent in enterprises. Employees’ appraisals and AI affect the smooth progress of AI–employee cooperation. This paper studies (1) whether employees’ challenge appraisals, threat appraisals and trust in AI are different for AI transparency and opacity. (2) This study investigates how AI transparency affects employees’ trust in AI through employee appraisals (challenge and threat appraisals), and (3) whether and how employees’ domain knowledge about AI moderates the relationship between AI transparency and appraisals. A total of 375 participants with work experience were recruited for an online hypothetical scenario experiment. The results showed that AI transparency (vs. opacity) led to higher challenge appraisals and trust and lower threat appraisals. However, in both AI transparency and opacity, employees believed that AI decisions brought more challenges than threats. In addition, we found the parallel mediating effect of challenge appraisals and threat appraisals. AI transparency promotes employees’ trust in AI by increasing employees’ challenge appraisals and reducing employees’ threat appraisals. Finally, employees’ domain knowledge about AI moderated the relationship between AI transparency and appraisals. Specifically, domain knowledge negatively moderated the positive effect of AI transparency on challenge appraisals, and domain knowledge positively moderated the negative effect of AI transparency on threat appraisals.

## 1. Introduction

Humans’ trust in artificial intelligence (AI) is helpful for the smooth progress of human–AI collaboration in enterprises because it can alleviate humans’ perceived risk of AI to a certain extent [1]. At present, people generally suspect that the existing AI is immature [2] and has moral and ethical issues [3,4], which seriously affect humans’ trust in AI. Transparency is an important feature of AI that can promote the development of humans’ trust in AI [1]. This is because the behavior of AI is not deterministic [5], and the decision-making process of AI is complex, multi-layered and opaque [1]. AI transparency refers to the degree to which an AI system releases information about its operation [6]. AI provides reasons or explanations for its own decisions, can greatly enhance employees’ understanding of the AI decision-making process and weaken the uncertainty of the results of AI decision-making [7].

However, previous studies have disputed whether AI transparency can improve humans’ trust in AI, and found that there may be a positive correlation, non-correlation, or inverted U-shaped relationship between AI transparency and trust [8]: (1) Positive correlation. AI transparency positively affected trust in automated leadership agents [9]. (2) Non-correlation. Cramer et al. [10] found that system transparency did not improve humans’ trust in an artwork recommendation system. (3) Inverted U-shaped relationship. Zhao et al. [6] claimed that objective transparency positively affected subjective transparency and that subjective transparency had an inverted U-shaped relationship with users’ understanding of systems, thereby affecting users’ trust in advice-giving systems. That is, providing too much detailed information reduces the user’s perceived understanding of advice-giving systems, thereby hurting the users’ trust [6].

This paper believes that the inconsistent relationship between AI transparency and trust may be that different people have different appraisals of AI. Firstly, different levels of transparency may produce different appraisals, thereby affecting trust. Dogruel [11] conducted an online experiment with Facebook users and found that compared with high-detailed advertising explanations, medium-detailed advertising explanations obtained more favorable evaluations among users. This author hinted that there might be a relationship between transparency and appraisals. People’s appraisals of AI may consist of challenging and threat appraisals. Because people will appraise the gains or losses of new technologies [12], challenge and threat appraisals concern losses and gains in an encounter [13]. The results of people’s appraisals of AI can be measured by the trust. Challenge and threat appraisals reflect employees’ perceptions of the extent to which AI can provide benefits or pose threats to them, which can help predict employees’ subsequent responses to AI [14]. Trust is often used as a predictor of technology acceptance [9,15].

Secondly, people with different domain knowledge regarding AI may have different appraisals of AI. Allen and Choudhury [16] found that workers with more domain experience had more aversion to algorithmic advice. Some examples in the literature suggest that there may be differences in the attitudes of experts and non-experts toward AI, and it might be interesting for future research to measure professional knowledge [17]. Therefore, this study believes that the appraisals of AI may be different among people with different domain knowledge, and domain knowledge may be a potential moderator of the relationship between AI transparency and appraisals.

In order to study the above issues, this paper built a theoretical model (see Figure 1) to study (1) how AI transparency affected employees’ trust in AI through employees’ challenge and threat appraisals. (2) Whether employees’ appraisals (challenge and threat appraisals) and employees’ trust in AI experienced differences between AI transparency and opacity. (3) Whether and how employees’ domain knowledge about AI moderated the relationship between AI transparency and appraisals (challenge and threat appraisals).

## 2. Theoretical Background and Research Hypotheses

### 2.1. Cognitive Appraisal Theory

Cognitive appraisal is the process of classifying various aspects of things encountered according to the meaning of happiness [13]. Lazarus and Folkman [13] believe that individuals have two kinds of cognitive appraisals when facing stress: primary appraisal and secondary appraisal. Primary appraisal refers to individuals’ judgments of the impact of the event on themselves, individuals who generate challenge appraisals focus on the gains or growth from an event, and individuals who generate threat appraisals focus on potential harm [13].

Previous studies have applied cognitive appraisal theory to AI research. Chiu et al. [18] used the cognitive appraisal theory to explore how appraisal factors affect employees’ behavioral responses through their affective and cognitive attitudes. Cao and Yao [14] studied the mediating role of opportunity and threat appraisals between AI functions and work outcomes. According to cognitive appraisal theory, challenge and threat appraisals can occur simultaneously and must be considered independently [13]. Therefore, challenge and threat appraisals are two variables. In this study, AI decision-making was used as a source of stress. Employees generate challenge and threat appraisals for AI transparency or opacity; trust is the result of the challenge and threat appraisals. It corresponds to the primary appraisal stage of the cognitive appraisal theory.

### 2.2. AI Transparency and Employees’ Trust in AI

Transparency is a way to increase trust [19]. For AI, important aspects of transparency include various types of explanations about how AI works or why it makes a particular decision, with explanations that can be understood by users even if they have no technical knowledge [1]. The positive correlation between AI system transparency and trust has been confirmed by previous empirical studies [9,20]. This study proposes that this positive correlation can also be realized in human-AI collaborative work scenarios. This is because, in many enterprises, the integration of AI into workflow requires AI to cooperate well with employees. AI transparency can increase employees’ trust in AI, make AI integrate better into the team, and improve the efficiency of human-AI collaboration [21].

AI transparency refers to the provision of reasons or explanations for AI’s decision-making [22]. This study follows earlier studies by defining AI opacity as simply announcing the final prediction results of the AI system to employees and AI transparency as announcing the final prediction results of the AI system to employees, along with information about the decision-making process [23]. The main difference between AI transparency and opacity is that transparency provides explanations for the decision-making process. Compared with no explanations, providing explanations leads to more trust [24]. Transparency shortens the knowledge gap between the AI system and employees by providing explanations. Through the transparent AI system’s explanations of decisions, employees could better understand the AI decision-making process and thus have more trust in the AI system. Therefore, we hypothesized that:

**Hypothesis 1** **(H1).**
*AI transparency makes employees have higher trust in AI than AI opacity.*


### 2.3. The Mediating Effect of Employees’ Challenge Appraisals

Challenge appraisals refer to the judgment of the gains or growth obtained in an event [13]. More explanations provided by the AI system can enable users to better understand the decision-making process of the AI system [6,25]. Compared with AI opacity, AI transparency provides more explanations for employees to understand their internal work. When employees can understand the internal working principles of AI, they build their belief in AI’s ability [26]. In other words, more transparent AI can be considered more capable, and this more capable AI enables employees to obtain more gains at work, and employees have higher challenge appraisals of AI. When employees obtain more benefits at work, it is beneficial to employees’ work outcomes [14,27]. When employees perceive the benefits of AI at work, they increase their trust in AI. For example, employees’ perceived usefulness of AI promotes employees’ trust in AI [28]. An employee’s perception of AI’s strong operational capabilities can improve trust in AI [18]. Therefore, we hypothesized that:

**Hypothesis 2** **(H2).**
*Employees’ challenge appraisals play a mediating role between AI transparency and employees’ trust in AI. AI transparency (vs. opacity) leads to higher employees’ challenge appraisals and, thereby, generates higher employees’ trust in AI.*


### 2.4. The Mediating Effect of Employees’ Threat Appraisals

Threat appraisals refer to the judgment of potential damage or losses in an event [13]. When AI decision-making is opaque, employees do not know how AI makes decisions, and they do not know AI’s decision-making process. Moreover, the environment in which AI runs is usually highly complex and has a certain degree of randomness, so the behavior of AI is not deterministic [5]. This leads employees to think that AI’s decision-making process is unpredictable and may trigger them to fear losing control at work, resulting in threat appraisals of AI [14]. However, the threat perception caused by AI can have a negative effect on employees’ work results and attitudes [29]. The increase in perceived threats among different groups reduces the group’s trust in external groups [30]. In this study, we believe that compared with AI opacity, AI transparency provides explanations that enable employees to understand the specific process of the AI system’s decision-making. These explanations can allow employees to reduce their sense of being out of control over AI and reduce their threat appraisals of AI, thereby enhancing trust. Therefore, we hypothesized that:

**Hypothesis 3** **(H3).**
*Employees’ threat appraisals play a mediating role between AI transparency and employees’ trust in AI. AI transparency (vs. opacity) leads to lower employee threat appraisals and, thereby, generates higher trust among employees in AI.*


### 2.5. Employees’ Appraisals of AI in AI Transparency and Opacity

In this study, AI transparency and opacity caused employees to have two judgments about the AI decision. According to Hypotheses 2 and 3, when AI was opaque, employees were afraid that AI would lose control. An opaque AI creates issues on trust, safety, ethics and fairness [31,32,33]. AI opacity may make employees feel that losses are greater than benefits; that is, employees generate more threat appraisals than challenge appraisals. Conversely, employees feel more benefits when AI is transparent, such as preventing negative outcomes and achieving accountability [22]. AI transparency may cause employees to feel that the benefits outweigh the losses; that is, employees generate more challenge appraisals than threat appraisals. Therefore, we hypothesized that:

**Hypothesis 4** **(H4).**
*When AI is opaque, the threat appraisals generated by employees are greater than challenge appraisals; when AI is transparent, the challenge appraisals generated by employees are greater than threat appraisals.*


### 2.6. The Moderating Effect of Employees’ Domain Knowledge of AI

There is a general positive bias toward new technologies. For example, users may tend to overestimate the performance of automated decision aids [34]. Employees often believe that automated aids are more reliable, leading to excessive reliance on them [24]. This may be because people know little about new technologies and have unrealistic optimistic beliefs about the capabilities and functions of new technologies, leading people to believe that new technologies are credible [24]. One piece of evidence comes from Logg et al. [35], who found that when laypeople think that advice comes from an algorithm rather than a person, they are more willing to follow it. In contrast, experienced professionals rely less on algorithmic advice than laypeople [35]. AI as a new technology also generates the above phenomenon. We believe that people with low domain knowledge have more positive biases toward AI. They overestimate AI’s capabilities and underestimate the risks of AI. By contrast, people with high domain knowledge have a better understanding of AI’s capabilities, and they have fewer positive biases toward AI. People with high domain knowledge (vs. low domain knowledge) can more objectively and accurately estimate the gains of AI at work, and they (vs. low domain knowledge) have a higher risk assessment of AI. Therefore, we hypothesized that:

**Hypothesis 5** **(H5).**
*Employees’ domain knowledge about AI plays a moderating role between AI transparency and employees’ challenge appraisals. That is, for employees with high domain knowledge (vs. low domain knowledge), the increase in AI transparency leads to lower challenge appraisals.*


**Hypothesis 6** **(H6).**
*Employees’ domain knowledge about AI plays a moderating role between AI transparency and employees’ threat appraisals. That is, for employees with high domain knowledge (vs. low domain knowledge), the increase in AI transparency leads to higher threat appraisals.*


## 3. Methodology

We used a hypothetical scenario experiment, which was described in words, to test our hypotheses. Participants were randomly assigned to one of two experimental scenarios (AI transparency: opacity vs. transparency). The contents of the two experimental scenarios were the same, except for the part about AI transparency being manipulated (see Appendix A). Participants were randomly assigned to different experimental scenarios, which differed in the manipulated text part. It is a common practice in transparency/AI/algorithm scenario experiments [9,36,37].

### 3.1. Sample and Data Collection

The sample size was calculated in advance using G * Power. The effect size f was 0.25, the α err prob, was 0.05, power (1-β err prob) was 0.95, and the number of groups was 2, while one-way ANOVA showed that the total sample size was 210 [38].

We recruited 400 participants with work experience online from Credamo (similar to Mturk and papers based on the data collected by Credamo that had been accepted by journals [39,40,41]) to complete online experimental research. Aguinis et al. [42] reported a range of attrition rates from 31.9% to 51% for MTurk research. We recruited 400 subjects based on this range. Through the online link, participants were randomly assigned to one of two scenarios: AI transparency or AI opacity. We conducted a strict screening of the questionnaire and strictly eliminated those that failed to pass the screening questions, whose answers were regular, and whose filling time was too short. Only 375 people (182 males vs.193 females; 191 opacities vs. 184 transparency) validly completed the experiment.

The characteristics of the sample are as follows: all participants were older than 18 years old, and most participants were between 25 and 35 years old (M = 31.51, SD = 5.39). The monthly income of most participants was between 8000 and 10,000 CNY (M = 9768, SD = 3917.42). The highest degree of most participants was a bachelor’s degree (76%). All participants had work experience, and participants came from various industries, with the largest proportions being production/processing/manufacturing (29.60%) and IT/communication/electronics/Internet (29.07%).

### 3.2. Procedure and Manipulation

Participants received the link from Credamo, opened the link and entered the experiment. First, participants were shown the informed consent page, on which we stated that the survey was anonymous and the information was kept confidential. After that, participants were randomly assigned to either AI transparency or opacity (see Appendix A).

Experimental scenarios described how, in a beverage company, a Sales Executive of the marketing management department worked with the AI system to make weekly sales forecasts. The experimental scenario with AI opacity simply announced the final prediction results of the AI system to employees, and the experimental scenario with AI transparency announced the final prediction results of the AI system to employees, along with information about the decision-making process. After reading the text, participants were asked to answer a questionnaire to measure the participants’ perceived transparency, challenge appraisals, threat appraisals, trust, domain knowledge scale, and finally, to fill in the demographic information.

### 3.3. Measures

In this study, we measured the construct “perceived transparency” to verify whether the manipulation of AI transparency was successful. Therefore, this experiment involved five constructs of: “perceived transparency”, “challenge appraisals”, “threat appraisals”, “trust”, and “domain knowledge”. All constructs were measured on 7-point Likert scales ranging from very inconsistent (1) to very consistent (7). The scale of perceived transparency came from Zhao et al. [6], the scale of challenge appraisals and threat appraisals came from Drach-Zahavya and Erez [43], the scale of trust came from Höddinghaus et al. [9], and the scale of domain knowledge came from Zhou et al. [44].

Since the experiment was carried out in China, we adopted back-translation techniques to translate the scales. According to the scenario, appropriate modifications were made, and the inverse questions were removed. For example, the original text of the domain knowledge item was “I know pretty much about jackets/digital cameras” before it was changed to “I know pretty much about AI systems”, and reverse items, such as “I do not feel very knowledgeable about jackets/digital cameras”, were deleted: “When it comes to jackets/digital cameras, I really don’t know a lot”. The measurement items of each construct are shown in Table 1.

## 4. Results

### 4.1. Validity and Reliability

SPSS 23 was used to test data reliability. The Cronbach’s α of perceived transparency, challenge appraisals, threat appraisals, trust, and domain knowledge were 0.969, 0.831, 0.888, 0.934, 0.899. Then, LISREL 8.80 was used to test data validity. The CFA results showed that the five-factor (trust, challenge appraisals, threat appraisals, perceived transparency, domain knowledge) model was a good fit according to the following fitting statistics: χ^2^ = 320.277, df = 109, χ^2^/df = 2.938, RMSEA = 0.072, NNFI = 0.968, CFI = 0.974, IFI = 0.974, GFI = 0.908, AGFI = 0.872. Moreover, the Harman single-factor model showed that the common method bias was not serious according to the following fitting statistics: χ^2^ = 3886.740, df = 119, χ^2^/df = 32.662, RMSEA = 0.291, NNFI = 0.471, CFI = 0.537, IFI = 0.538, GFI = 0.450, AGFI = 0.293. Only one item of standardized factor loading was less than 0.5, which was 0.490. The other items were between 0.655 and 0.972, and all reached a high level of significance (*p* < 0.001), showing adequate convergence validity. The Cronbach’s α of each construct was greater than 0.8, the combined reliability (CR) was greater than 0.8, and the average variance extracted (AVE) was greater than 0.50, which are all acceptable. Table 2 shows the results of the reliability and validity analysis.

In addition, Table 3 shows the mean value, standard deviation, correlation coefficients and square root AVE of each construct in this study. All constructs are significantly correlated. The square root of the AVE for each construct was between 0.754 and 0.956, which is greater than the correlation coefficient between the constructs, indicating adequate discriminative validity.

### 4.2. Test the Manipulation of AI Transparency

In order to test whether the manipulation of the AI transparency and opacity was successful, we introduced employees’ perceived transparency to verify. The results of the one-way ANOVA showed that the mean value of perceived transparency in the transparency group was significantly higher than that in the opacity group (F (1, 373) = 1589.495, *p* < 0.001, M_opacity_ = 2.007 < M_transparency_ = 5.944), indicating that this study successfully manipulated AI transparency.

### 4.3. Difference Test for Challenge Appraisals, Threat Appraisals and Trust

As shown in Table 4, the results of the one-way ANOVA showed that the mean value of trust in the transparency group was significantly higher than that in the opacity group (F (1, 373) = 133.063, *p* < 0.001, M_opacity_ = 3.967 < M_transparency_ = 5.476). Therefore, H1 was supported.

As shown in Table 4, the mean value of challenge appraisals in the transparency group was significantly higher than that in the opacity group (F (1, 373) = 23.687, *p* < 0.001, M_opacity_ = 4.795 < M_transparency_ = 5.344). The mean value of threat appraisals in the transparency group was significantly lower than that in the opacity group (F (1, 373) = 4.281, *p* < 0.05, M_opacity_ = 2.916 > M_transparency_ = 2.650). This lays the foundation for the verification of the mediation effect.

In order to compare the mean value of challenge appraisals and threat appraisals, the paired sample T-test was conducted. As shown in Table 5, in both AI opacity and transparency groups, the mean value of challenge appraisals was greater than that of the threat appraisals. Therefore, H4 was partially supported.

### 4.4. Test of Mediating Effect

In order to test the mediating effect of the challenge and threat appraisals, model 4 of the SPSS PROCESS Macro developed by Hayes [45] was used for regression analysis (see Table 6). Trust was used as the dependent variable, AI transparency (0 = opacity, 1 = transparency) was used as the independent variable, challenge appraisals and threat appraisals were used as mediating variables, and gender, age, and educational background were used as the control variables. The results showed that AI transparency had a significant positive effect on trust (β = 1.528, *p* < 0.001). After adding the challenge appraisals and threat appraisals as mediating variables, AI transparency still had a significant positive effect on trust (*p* = 1.283, *p* < 0.001). As shown in Table 7, the confidence interval of the mediating effect of challenge appraisals was [0.068, 0.272] and did not include 0; the confidence interval of the mediating effect of threat appraisals was [0.009, 0.183] and did not include 0. Therefore, challenge and threat appraisals played a partial mediating role between AI transparency and trust, while H2 and H3 were supported.

### 4.5. Test of Moderating Effect

To test the moderating effect of domain knowledge, model 7 of the SPSS PROCESS Macro developed by Hayes [45] was used for regression analysis (see Table 8). Trust was used as the dependent variable, AI transparency (0 = opacity, 1= transparency) was used as the independent variable, challenge appraisals and threat appraisals were used as mediating variables, domain knowledge was used as the moderating variable, and gender, age, and educational background were used as the control variables. The results showed that the cross-term of AI transparency and domain knowledge had a significantly negative effect on challenge appraisals (β = −0.201, *p* < 0.05), and H5 was supported. The cross-term of AI transparency and domain knowledge had a significantly positive effect on threat appraisal (β = 0.291, *p* < 0.05), and H6 was supported.

## 5. Discussion

This paper mainly studies three questions: first, how does AI transparency affect trust through challenge and threat appraisals; second, do challenge appraisals, threat appraisals and trust experience differences between AI transparency and opacity; third, does employees’ domain knowledge about AI moderate the relationship between AI transparency and appraisals (challenge and threat appraisals)? Through an empirical test, H1, H2, H3, H5 and H6 were supported, while H4 was only partially supported. These results are discussed below.

Firstly, AI transparency (vs. opacity) produces higher challenge appraisals and trust and lower threat appraisals. This result proves that AI transparency provides employees with higher positive appraisals of AI. In particular, this paper found that in both AI transparency and opacity scenarios, employees considered the challenges of AI decision-making to be greater than the threats. This result only supports part of the hypothesis of H4. The inconsistency is that we assumed that threat appraisals were greater than challenge appraisals in the AI opacity scenario. This may be because, although AI in enterprises is now mostly opaque [46], employees have become accustomed to the existence of AI and believe that AI is efficient; that is, opportunity appraisals are greater than threat appraisals.

Secondly, employees’ challenge appraisals and threat appraisals play a partial mediating role between AI transparency and employees’ trust in AI. AI transparency increases employees’ trust in AI by increasing employees’ challenge appraisals and reducing employees’ threat appraisals. This result is similar to Cao and Yao’s [14] finding that employees’ appraisals of AI affect their behavioral results.

Thirdly, employees’ domain knowledge about AI moderates the relationship between AI transparency and appraisals (challenge and threat appraisals). Domain knowledge negatively moderates the positive effect of AI transparency on challenge appraisals, and domain knowledge positively moderates the negative effect of AI transparency on threat appraisals. This result suggests that employees with high domain knowledge (vs. employees with low domain knowledge) tend to view AI transparency more rationally, and they (vs. employees with low domain knowledge) do not overestimate the benefits of transparent AI and do not underestimate the losses of transparent AI. This was also proved by the discovery of Gutiérrez et al. [47]; namely, for the prediction and advice of algorithms, laymen believe that algorithms are more accurate and effective than experts.

### 5.1. Theoretical Implications

The theoretical implications of this study are as follows: first, the previous literature has studied the influence of different degrees of AI transparency on trust [7,48] but there are relatively few studies on whether different degrees of AI transparency affect the appraisals of AI. Focusing on employee–AI collaboration scenarios, this paper proves that in an AI transparency (vs. opacity) scenario, employees generate higher challenge appraisals and trust, and lower threat appraisals. In both AI transparency and opacity scenario, employees believe that the challenges brought by AI decision-making outweigh the threats. In addition, this paper found that employees’ challenges and threat appraisals of AI had a parallel mediating effect between AI transparency and employees’ trust in AI.

Second, predecessors have found differences in people’s attitudes towards AI from the perspective of domain knowledge [16,35], but there are few studies on whether domain knowledge affect the appraisals of AI under the stimulation of transparency. At the same time, because AI has just been introduced into enterprises, some employees do not understand AI at all, while others know it well, resulting in great individual differences among employees. Enterprises want to increase employees’ domain knowledge concerning AI through a series of measures such as education and training; that is, employees’ domain knowledge about AI is a variable that enterprises can manipulate. Therefore, this paper takes into account the heterogeneity of employees and takes employees’ domain knowledge about AI to be a moderating variable and finds that domain knowledge moderates the relationship between AI transparency and challenge appraisals and AI transparency and threat appraisals.

Third, few existing AI studies use cognitive appraisal theory, and there is a lack of evidence in the literature on whether employees generate both challenge appraisals and threat appraisals when AI transparency is used as a source of stress. This study extends the application of the cognitive appraisal theory to employees’ appraisals of AI transparency in enterprises and confirms that employees evaluate (calculate gains and losses) in the face of AI transparency and opacity, thus affecting their trust in AI.

### 5.2. Practical Implications

According to the research results of this paper, we believe that the practical implications are mainly in the following three areas: first, employees working with AI believe that AI brings more challenges than threats. In the future, enterprises can rely on AI to make decision agents for part of their daily work. Coleman Parks Research explored the perception of AI among hourly and salaried workers in multiple countries and found that four out of five employees could see the potential benefits of AI for improving the workplace experience [49].

Second, this paper found that AI transparency positively affects employees’ trust in AI, both directly and by increasing employees’ challenge appraisals and reducing employees’ threat appraisals. Therefore, enterprises should improve the transparency of AI systems, such as informing employees of the process of AI decision-making or designing more transparent AI systems. IBM offers a business AI platform product with prominent functions of “trust, transparency and interpretability” because the product provides tools to help explain and manage AI-led decisions in the enterprise [50]. When employees collaborate with AI, it is important to make them understand the gains of the work and reduce their sense of being out of control of the work so as to increase their challenge appraisals and reduce their threat appraisals. 

Third, we found that employees with low domain knowledge (vs. high domain knowledge) concerning AI have higher positive appraisals of AI. However, these positive appraisals come from a blind overestimation. Employees with low domain knowledge regarding AI overestimate the capabilities of AI and underestimate the risks of AI. In the short term, this is conducive to collaboration with AI, but in the long term, is not conducive to the development of human–AI collaboration. Therefore, the risks brought by AI collaboration should be correctly assessed, and employees should be educated about AI.

### 5.3. Limitations and Suggestions for Future Research

The research of this paper has the following limitations. First of all, this study used a hypothetical scenario experiment. Participants were asked to imagine themselves in the described scenario by reading a paragraph of text. Participants may not be able to personally experience this scenario, which may have resulted in them filling out the questionnaire based solely on their own experience. Future research should further refine this scenario so that participants can experience a real scenario. Secondly, the employees’ domain knowledge about AI came from self-evaluation, which was not objective. Future research should use more objective methods to test this variable. Finally, this study only conducted the experiment on a sale forecast scenario, and the results may be affected by the experimental scenario. Future research should repeat the experiment in different scenarios to test the stability of the model and results.

## Figures and Tables

**Figure 1 behavsci-13-00344-f001:**
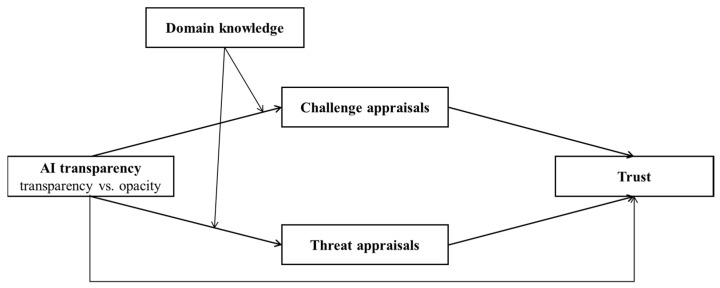
The theoretical model.

**Table 1 behavsci-13-00344-t001:** Measurement items of each construct.

Constructs	Items	References
Perceived transparency	I can access a great deal of information which explains how the AI system works.	Zhao et al. [6]
	I can see plenty of information about the AI system’s inner logic.	
	I feel that the amount of available information regarding the AI system’s reasoning is large.	
Challenge appraisals	AI-involved work seems like a challenge to me.	Drach-Zahavya and Erez [43]
	AI-involved work provides opportunities to exercise reasoning skills.	
	AI-involved work provides opportunities to overcome obstacles.	
	AI-involved work provides opportunities to strengthen my self-esteem.	
Threat appraisals	AI-involved work seems like a threat to me.	Drach-Zahavya and Erez [43]
	I’m worried that AI-involved work might reveal my weaknesses.	
	AI-involved work seems long and tiresome.	
	I’m worried that AI-involved work might threaten my self-esteem.	
Trust	I would heavily rely on the AI system.	Höddinghaus et al. [9]
	I would trust the AI system completely.	
	I would feel comfortable relying on the AI system.	
Domain knowledge	I know pretty much about AI systems.	Zhou et al. [44]
	Among my circle of friends, I’m one of the “experts” on AI systems.	
	Compared to most other people, I know less about AI systems.	

**Table 2 behavsci-13-00344-t002:** Results of reliability and validity analysis (N = 375).

Constructs	Items	Standardized Factor Loading (λ)	*t*-Value	Cronbach’s α	CR	AVE
Trust	TRU01	0.922	23.072	0.934	0.936	0.829
	TRU02	0.891	21.788			
	TRU03	0.918	22.903			
Challenge appraisals	CHA01	0.490	9.523	0.831	0.835	0.568
	CHA02	0.817	18.271			
	CHA03	0.860	19.701			
	CHA04	0.791	17.460			
Threat appraisals	THR01	0.853	20.007	0.888	0.892	0.676
	THR02	0.866	20.451			
	THR03	0.655	13.777			
	THR04	0.892	21.426			
Perceived transparency	PER01	0.972	25.735	0.969	0.969	0.913
	PER02	0.953	24.825			
	PER03	0.942	24.292			
Domain knowledge	KNO01	0.846	19.650	0.899	0.902	0.755
	KNO02	0.888	21.163			
	KNO03	0.872	20.581			

**Table 3 behavsci-13-00344-t003:** The matrix of correlation coefficients.

Constructs	Mean	SD	1	2	3	4	5
1. Trust	4.708	1.474	0.910				
2. Challenge appraisals	5.064	1.125	0.495 ***	0.754			
3. Threat appraisals	2.785	1.254	−0.385 ***	−0.389 ***	0.822		
4. Perceived Transparency	3.939	2.190	0.596 ***	0.395 ***	−0.145 **	0.956	
5. Domain knowledge	4.908	1.184	0.414 ***	0.551 ***	−0.370 ***	0.351 ***	0.869

Note: The data on the diagonal line are the square root of AVE, and the data on the off-diagonal line are the correlation coefficient among latent constructs; ** means *p* < 0.01, *** means *p* < 0.001.

**Table 4 behavsci-13-00344-t004:** Results of the one-way ANOVA.

	AI Opacity	AI Transparency	
Challenge appraisals	4.795 (SE = 0.091)	5.344 (SE = 0.065)	F (1, 373) = 23.687, *p* < 0.001
Threat appraisals	2.916 (SE = 0.094)	2.650 (SE = 0.088)	F (1, 373) = 4.281, *p* < 0.05
Trust	3.967 (SE = 0.116)	5.476 (SE = 0.058)	F (1, 373) = 133.063, *p* < 0.001

**Table 5 behavsci-13-00344-t005:** Results of the paired sample *t*-test.

	Correlation	Challenge Appraisals	ThreatAppraisals	*t*-Value	df	Sig
AI opacity	−0.322 ***	4.795 (SE = 0.091)	2.916 (SE = 0.094)	12.465	190	*p* < 0.001
AI transparency	−0.242 ***	5.344 (SE = 0.065)	2.650 (SE = 0.088)	22.185	183	*p* < 0.001

Note: *** means *p* < 0.001.

**Table 6 behavsci-13-00344-t006:** The results of the mediating effect.

Dependent Variable	Independent Variable	β	SE	T	95% Confidence Interval		R^2^	F
					LLCI	ULCI		
Trust	Constant	3.677 ***	0.568	6.472	2.560	4.794	0.265	33.304 ***
	AI transparency	1.528 ***	0.134	11.407	1.265	1.792		
	Gender	−0.100	0.133	−0.753	−0.360	0.161		
	Age	0.008	0.013	0.633	−0.017	0.033		
	Educational background	0.026	0.125	0.211	−0.219	0.271		
Challenge appraisals	Constant	4.623 ***	0.489	9.448	3.661	5.585	0.064	6.356 ***
	AI transparency	0.531 ***	0.115	4.603	0.304	0.758		
	Gender	−0.016	0.114	−0.137	−0.240	0.209		
	Age	−0.006	0.011	−0.577	−0.027	0.015		
	Educational background	0.126	0.107	1.175	−0.085	0.337		
Threat appraisals	Constant	4.399 ***	0.553	7.954	3.311	5.486	0.037	3.558 **
	AI transparency	−0.284 *	0.130	−2.174	−0.540	−0.027		
	Gender	0.101	0.129	0.784	−0.153	0.355		
	Age	−0.015	0.012	−1.226	−0.039	0.009		
	Educational background	−0.346 **	0.121	−2.850	−0.585	−0.107		
Trust	Constant	3.690 ***	0.639	5.773	2.433	4.947	0.413	43.056 ***
	AI transparency	1.283 ***	0.124	10.383	1.040	1.526		
	Challenge appraisals	0.294 ***	0.057	5.214	0.183	0.406		
	Threat appraisals	−0.312 ***	0.050	−6.253	−0.411	−0.214		
	Gender	−0.064	0.119	−0.535	−0.297	0.170		
	Age	0.005	0.011	0.452	−0.017	0.027		
	Educational background	−0.119	0.113	−1.053	−0.341	0.103		

Note: * means *p* < 0.05, ** means *p* < 0.01, *** means *p* < 0.001.

**Table 7 behavsci-13-00344-t007:** Total effect, direct effect and mediating effect.

		Effect	SE	95% Confidence Interval	
				LLCI	ULCI
Total effect		1.528	0.134	1.265	1.792
Direct effect		1.283	0.124	1.040	1.526
Mediating effect	Total	0.245	0.073	0.112	0.399
	Challenge appraisals	0.156	0.052	0.068	0.272
	Threat appraisals	0.089	0.044	0.009	0.183

**Table 8 behavsci-13-00344-t008:** The results of the moderating effect.

Dependent Variable	Independent Variable	β	SE	T	95% Confidence Interval		R^2^	F
					LLCI	ULCI		
Challenge appraisals	Constant	2.861 ***	0.497	5.760	1.885	3.838	0.244	19.785 ***
	AI transparency	1.268 *	0.491	2.584	0.303	2.233		
	Domain knowledge	0.485 ***	0.056	8.610	0.374	0.596		
	AI transparency × Domain knowledge	−0.201 *	0.095	−2.116	−0.389	−0.014		
	Gender	−0.063	0.104	−0.604	−0.267	0.141		
	Age	−0.007	0.010	−0.719	−0.026	0.012		
	Educational background	−0.010	0.098	−0.101	−0.202	0.183		
Threat appraisals	Constant	6.048 ***	0.590	10.257	4.889	7.207	0.142	10.139 ***
	AI transparency	−1.526 **	0.583	−2.619	−2.671	−0.380		
	Domain knowledge	−0.436 ***	0.067	−6.522	−0.568	−0.305		
	AI transparency × Domain knowledge	0.291 *	0.113	2.577	0.069	0.514		
	Gender	0.124	0.123	1.003	−0.119	0.366		
	Age	−0.015	0.012	−1.317	−0.038	0.008		
	Educational background	−0.231 *	0.116	−1.991	−0.460	−0.003		
Trust	Constant	3.690 ***	0.639	5.773	2.433	4.947	0.413	43.056 ***
	AI transparency	1.283 ***	0.124	10.383	1.040	1.526		
	Challenge appraisals	0.294 ***	0.057	5.214	0.183	0.406		
	Threat appraisals	−0.312 ***	0.050	−6.253	−0.411	−0.214		
	Gender	−0.064	0.119	−0.535	−0.297	0.170		
	Age	0.005	0.011	0.452	−0.017	0.027		
	Educational background	−0.119	0.113	−1.053	−0.341	0.103		

Note: * means *p* < 0.05, ** means *p* < 0.01, *** means *p* < 0.001.

## Data Availability

Not applicable.

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
