# Peer review of "Employees’ Appraisals and Trust of Artificial Intelligences’ Transparency and Opacity"

_behavsci, 2023, doi:10.3390/bs13040344_

Round 1
Reviewer 1 Report
The article is related to the trust in artificial intelligence in employees.
It is considered a topical and interesting subject, and the article adequately meets the requirements of a quality scientific research. Both the introduction, the hypothesis statement, the methodology, the results and the subsequent discussion are suitable.
The only thing I would add is a section in the methodology referring to the research design, before the sample and data collection, in which it is specified more clearly what the experiment will consist of, it is indicated that previous studies have been carried out with this research method and the research approach is adequately and more clearly stated.
Author Response
Response to Reviewer 1 Comments
Point 1: The only thing I would add is a section in the methodology referring to the research design, before the sample and data collection, in which it is specified more clearly what the experiment will consist of, it is indicated that previous studies have been carried out with this research method and the research approach is adequately and more clearly stated.
Response 1: Thank you for pointing this out to us. We have added references to the research design in Methodology. Methodology has revised as follows:
3. Methodology
We used the hypothetical scenario experiment described in words to test our hypotheses. Participants were randomly assigned to one of two experimental scenarios (AI transparency: opacity vs. transparency). The contents of the two experimental scenarios are the same, except for the part of AI transparency being manipulated (see Appendix A). Participants were randomly assigned to different experimental scenarios, which differed in the manipulated text part. It is a common practice in transparency/AI/algorithm scenario experiment [9,36,37].
Reviewer 3 Report
Thanks for submitting this interesting study. Certainly, this topic is worth further research. Nevertheless, only one thing worried me whilst reviewing your papers: Your methodology, sample, and data collection section lack referencing, and I wondered why the referencing is missing.
Author Response
Response to Reviewer 3 Comments
Point 1: Thanks for submitting this interesting study. Certainly, this topic is worth further research. Nevertheless, only one thing worried me whilst reviewing your papers: Your methodology, sample, and data collection section lack referencing, and I wondered why the referencing is missing.
Response 1: We apologize for the lack of references in methodology, sample and data collection. We have added references to methodology, sample and data collection. Methodology, Sample and data collection have revised as follows:
3. Methodology
We used the hypothetical scenario experiment described in words to test our hypotheses. Participants were randomly assigned to one of two experimental scenarios (AI transparency: opacity vs. transparency). The contents of the two experimental scenarios are the same, except for the part of AI transparency being manipulated (see Appendix A). Participants were randomly assigned to different experimental scenarios, which differed in the manipulated text part. It is a common practice in transparency/AI/algorithm scenario experiment [9,36,37].
3.1. Sample and data collection
The sample size was calculated in advance using G*Power. Effect size f is 0.25, α err prob is 0.05, power (1-β err prob) is 0.95, and the number of groups is 2, one-way ANOVA shows the total sample size is 210 [38].
We recruited 400 participants with work experience online from Credamo (similar to Mturk, and papers based on the data collected by Credamo have been accepted by journals [39-41]) to complete online experimental research. Aguinis et al. [42] reported a range of attrition rates from 31.9% to 51% for the MTurk research. We recruited 400 subjects based on this range. Through the online link, participants were randomly assigned to one of two scenarios, with AI transparency or AI opacity. We conducted a strict screening of the questionnaire, and strictly eliminated those that failed to pass the screening questions, whose answers were regular and whose filling time was too short. Only 375 people (182 males vs.193 females; 191 opacity vs. 184 transparency) validly completed the experiment.
The characteristics of the sample are as follows: all participants are older than 18 years old, and most participants are between 25 and 35 years old (M = 31.51, SD = 5.39). The monthly income of most participants is between 8000 and 10000 CNY (M=9768, SD=3917.42). The highest degree of most participants is a bachelor’s degree (76%). All participants have work experience, and participants come from various industries, with the largest proportions being production/processing/manufacturing (29.60%) and IT/communication/electronics/Internet (29.07%).
Round 2
Reviewer 2 Report
I think most of the comments are well addressed. Thank you for your effort.